

# Effects of turning over intensity on fatty acid metabolites in postharvest leaves of Tieguanyin oolong tea (*Camellia sinensis*)

Zixin Ni[1,*], Qingyang Wu[1,*], Ziwei Zhou[1,2], Yun Yang[1], Qingcai Hu[1], Huili Deng[1], Yucheng Zheng[1], Wanjun Bi[1], Zhenzhang Liu[1] and Yun Sun[1]

[1] College of Horticulture/Key Laboratory of Tea Science in Fujian Province, Fujian Agriculture and Forestry University, Fuzhou, China
[2] College of Life Science, Ningde Normal University, Ningde, China
[*] These authors contributed equally to this work.

## ABSTRACT

Fatty acid derived volatiles (FADVs) are major contributors to the aroma quality of oolong tea (*Camellia sinensis*). Most of the processing time for oolong tea is taken up by turning over treatments, but the full profile of fatty acid metabolic changes during this process remains unclear. In this study, we detected fatty acids, their derived volatiles, and related genes of Tieguanyin oolong tea using biochemical and molecular biology methods. The results showed that with an increase in turning over intensities, the content of total unsaturated fatty acids continuously dropped and the content of characteristic FADVs, such as hexanoic acid (Z)-3-Hexenly ester and 2-exenal, continued to increase. Lipoxygenase (LOX), a key gene family in the fatty acid metabolic pathway, showed different patterns, and *CsLOX1* (TEA025499.1) was considered to be a key gene during the turning over processes. We found that fruit-like aroma (Z)-3-Hexen-1-ol acetate had a strong correlation with the expression levels of eight *Camelia sinensis LOX* family genes. Tieguanyin had relatively rich pleasant volatile compounds with moderate turning over intensity (five times turning over treatments). This study provides an overall view of how fatty acid metabolites change and affect the quality of oolong tea with different turning over intensities during processing.

## INTRODUCTION

Oolong tea, a semi-oxidized tea made from *Camellia sinensis*, has abundant floral and fruity aroma volatiles and a taste that is considered to be between black and green tea (*Zeng et al., 2017*; *Ng et al., 2018*). It is considered an important functional food due to its secondary metabolites (*Weerawatanakorn et al., 2015*). The process of making oolong tea includes several procedures such as plucking, withering, turning over, fixing, rolling, and drying fresh leaves (*Hara et al., 1995*; *Zeng et al., 2020*). Turning over is the key process where post-harvest leaves are turned over which causes continuous mechanical damage and is the major cause for oolong tea's unique aroma (*Hu et al., 2018*; *Lin et al., 2013*).

Corresponding author
Yun Sun, sunyun1125@126.com

The corresponding relationship between turning over techniques and tea quality has been widely investigated, and different turning over treatments create great differences in tea aroma and taste factors (*Zhou et al., 2017a*; *Zhou et al., 2017b*; *Zhu et al., 2013*; *Huang et al., 2003*). As turning over intensity increases, the content of water extract, polyphenols, catechin, and soluble sugars decreases (*Huang et al., 2002*), but aroma components such as esters, ketones, aldehydes, typical floral volatiles nerolidol, linalool, and $\alpha$-farnesene accumulate (*Katsuno et al., 2014*; *Yang, Baldermann & Watanabe, 2013*).

Lipoxygenase (LOX) is a key enzyme in the fatty acid metabolic pathway that can oxidize fatty acids (FAs) and is involved in the generation of a series of volatiles and hormones under both abiotic and biotic stresses such as low temperature, drought, mechanical stress, and pathogens (*Upchurch, 2008*; *Liavonchanka & Feussner, 2006*; *Grechkin, 1998*). Fatty acid derived green leaf volatiles and jasmonates were shown to regulate wound-associated defenses, and *LOX10* may be involved in wound-induced oxylipin responses (*He et al., 2020*). *LOX* gene expression and enzyme activities have also been reported to have strong correlations with flavor and aroma formation in kiwifruit (*Zhang et al., 2009*), melon (*Tang et al., 2015*), strawberry (*Leone et al., 2006*), and apple (*Defilippi, Dandekar & Kader, 2005*). The fatty acid metabolic pathway has recently received attention for its important role in aroma formation's contribution to tea quality. Fatty acid derived compounds such as hexanal, (Z)-3-hexen-1-ol, and hexanoic acid formed esters with fruit-like aromas in tea production (*Zhou et al., 2021a*). During the withering process in black tea manufacturing, the content of FAs dropped rapidly (*Pradip, Pradip & Lakshi, 1993*). In contrast to leaves without mechanical stress, leaves under mechanical stress showed a significant drop in the content of linolenic acid (*Zhou et al., 2019*). A-linolenic and linoleic acid form $C_6$ compound volatiles during oolong tea processing (*Zhou et al., 2020*). During oolong tea processing, unsaturated fatty acids (UFAs) are used as reaction substrates to produce $C_6$ or $C_9$ alcohols, aldehydes, acids, and their derived esters with floral and fruity aroma characteristics under the action of enzymes involving LOX (*Revichandran & Parthiban, 2000*). Its activity increased in the early stages of withering and turning over, and it was assumed to lead to massive ester aroma formation (*Wang, 2010*). *CsLOX1* in oolong tea was triggered and then expressed after mechanical stress, leading to the accumulation of jasmine lactone, which has a fruity and floral odor (*Zeng et al., 2018*).

Until now, the whole profile of fatty acid substrates, fatty acid derived volatiles (FADVs), *CsLOX* gene expression levels, and their correlation in the fatty acid metabolic pathway during the turning over manufacturing process in oolong tea has yet to be determined. In this study, we detected and analyzed the change patterns of FAs and volatile compounds during oolong tea's turning over process using a gas chromatography-flame ionization detector (GC-FID) and head space-solid phase micro extraction-gas chromatograph-mass spectrometer(HS-SPME-GC-MS). Additionally, the expression patterns of *CsLOX* gene family members varied across different turning over intensities, and the correlation analyses among certain *CsLOX* genes, FAs, and fatty acid derived aroma were also performed. Our results provide an overall view of the fatty acid metabolic pathway during the key process of oolong tea manufacturing, and can also help develop oolong tea processing techniques and improve tea quality.

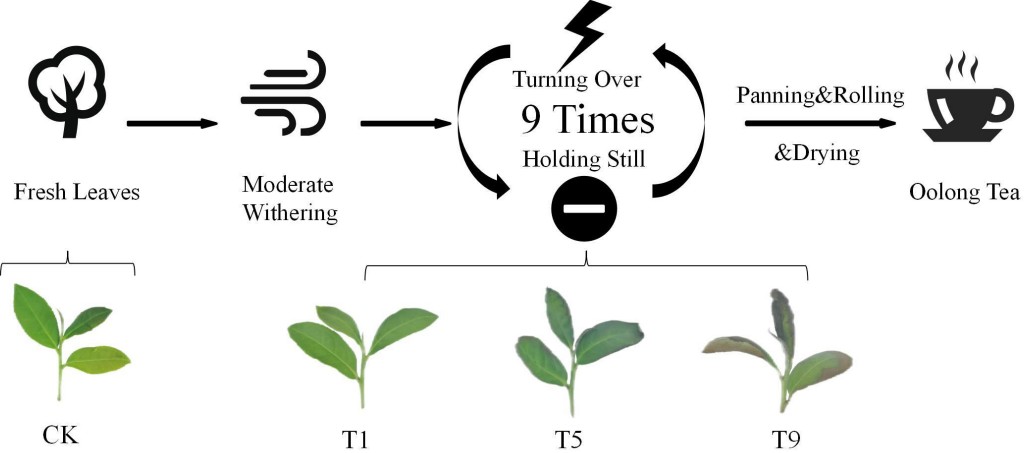

**Figure 1** **Sampling points during the Tieguanyin oolong tea manufacturing process.** CK, fresh leaves plucked from the Tieguanyin tea tree. Turning over intensities increased with the turning over times. Each turning over treatment was followed by 30 min standing time; T1, T5, and T9 were samples following the first, fifth, and ninth turning over treatments, respectively.

## MATERIALS & METHODS

### Plant materials and treatments

We collected one bud and three leaves from *Camellia cinensis* Tieguanyin, one of the most popular tea cultivar used for processing oolong tea, from Fuzhou Migaoxian Tea Industry Development Co., Ltd. (26°18′N, 119°39′E, Fuzhou, China) in October 2019. Based on the morphology of tea leaves following moderate withering, nine rounds of turning over experiments were carried out with a turnover machine for 1 min, 2 min, 3 min, 4 min, 5 min, 6 min, 7 min, 8 min, and 9 min, respectively, at a rate of 30 rounds per min, with 30 min standing time between every two turning over experiments. Samples were collected after plucking (CK), the first turning over experiment (T1), the fifth turning over experiment (T5), and the ninth turning over experiment (T9) (Fig. 1). T1, T5, and T9 were regarded as the representatives of slight, moderate, and heavy turning over, respectively. All sampled leaves were frozen using liquid nitrogen and stored in a −70 °C refrigerator for further use.

### Fatty acid determination

We weighed 100 mg (freeze-dried weight) of finely ground tea sample and added 50 μL of C17:0 (internal standard), two mL of 5% concentrated sulfuric acid/methanol solution, and 300 μL of toluene to a container sealed with an aluminum bottle cap and a poly tetra fluoroethylene silicone pad. After shaking gently, the mixed sample was extracted in a 92 °C constant temperature water bath for 1.5 h. After extraction and cooling to room temperature, one mL of 0.9% NaCl was added to the sample, and it was shaken gently. Subsequently, one mL of n-hexane was added for extraction and the sample was centrifuged for 3 min at 4,000 rpm. The supernatant was then taken and added to the injection bottle for further detection. We used a gas chromatography-flame ionization

detector (GC-FID) and an Agilent 7890A gas chromatograph mass spectrometer (Agilent Technologies, Santa Clara, CA, USA) with a HP-FFAP column (30 m × 0.25 mm × 0.25 μm) and a flame ionization detector (FID). Split injection was used, and the ratio was 20:1. The temperature of the injection port and detector was set at 250 °C and 260 °C, respectively. The temperature rise procedure was as follows: the initial temperature was 80 °C, maintained for 30 s, then rose to 165 °C at the rate of 40 °C min$^{-1}$, and lasted for 1 min. Finally, the temperature rose to 230 °C at a rate of 4 °C min$^{-1}$ and was maintained for 6 min. The FAs were detected and the content was measured using retention time, the peak area of detected FAs (S1), the peak area of internal standard (S2), the content of internal standard (N), and the mass of the sample (M) with the formula content (%) = (S1/S2) × (N/M) × 100%.

## Detection of volatile compounds using a head space—solid phase micro extraction—gas chromatograph-mass spectrometer (HS-SPME-GC-MS)

We placed 2.0 g (freeze-dried weight) of finely ground tea samples into 20 mL solid phase micro extraction (SPME) bottles that were sealed by caps with poly tetra fluoroethylene silicone pads. Head space—solid phase micro extraction—gas chromatograph-mass spectrometer (HS-SPME-GC-MS) detection was performed using an Agilent 7890B gas chromatograph mass spectrometer (Agilent Technologies, Santa Clara, CA, USA) equipped with an Rxi®-5silMS column (30 m × 0.25 mm × 0.25 μm) and a Pegasus HT time-of-flight mass spectrometer (Leco, St. Joseph, MI, USA). A PDMS/DVB (23 Ga, plain, 65 μm; Supelco, Bellefonte, PA, USA) syringe fiber was used and pre-baked for 30 min at 250 °C. The incubation temperature was set at 80 °C for 31 min, and the extraction of samples lasted for 60 min before 5 min desorption. The temperature of the injection port was set at 250 °C. High purity helium was used as the carrier gas at a flow of one mL min$^{-1}$. The initial temperature of the GC oven was 50 °C for 5 min, then ramped up to 210 °C at a rate of 3 °C min$^{-1}$, was maintained for 3 min, and finally rose to 230 °C at a rate of 15 °C min$^{-1}$. Ion source temperature was set at 250 °C and was generated by a −70 eV electron beam. The scan range was 30–500 atomic mass units, and the solvent delay time was 200 s at a rate of 10 spectra/s. The volatile compounds were determined using retention time and the national institute of standards and technology (NIST) database, and the content was determined using the area normalization method.

## RNA isolation, cDNA synthesis, and quantitative real-time PCR assay

Total RNA was extracted from the tea leaf sample using an RNAprep Pure Plant Kit (Tiangen Biotech Co. Ltd., Beijing, China) according to the manufacturer's instructions. The quality and concentration of isolated RNA was assessed using a spectrometer (Thermo Fisher, Waltham, MA, USA). Following that, cDNA was synthesized using a PrimeScript RT Reagent Kit with a gDNA Eraser (TaKaRa Biotech Co., Ltd., Dalian, China). The *CsLOX* gene family members were identified using the Pfam database and included lipoxygenase protein domain PF00305 and tea genome database TPIA (http://tpia.teaplant.org/index.html) (*Finn et al., 2008*; *Xia et al., 2019*). *CsLOX1 ~CsLOX10* were named according to their homologous sequence alignments in NCBI and previous

**Table 1  The primers of *CsLOX* genes and reference genes for qRT-PCR.**

| Gene name | Gene ID or accession number | Forward/reverse primer sequence (5′–3′) | Temprature (°C) |
|---|---|---|---|
| *CsLOX1* | TEA025499.1 | CATCCACAAAGATTTACGCCAC/AAACGGAGCCTTCAACACC | 60 |
| *CsLOX2* | TEA009423.1 | TGATGGTCTTGTTCTCTGGGA/GCTCTTCGTCGGACTTTATGAA | 60 |
| *CsLOX3* | TEA011776.1 | TGAAGAGGCTACGAGAGGAAG/GGATGTTGTTTACCACCGAGTA | 58 |
| *CsLOX4* | TEA009596.1 | ATGGGATGAGGAAGTTGGG/CTGATTAGTGAAGAACACACGGTC | 58 |
| *CsLOX5* | TEA012289.1 | GTGAACCGTTACTACCCAGATT/TACTGTCCAAAGTTGAGGGC | 58 |
| *CsLOX6* | TEA020304.1 | ACTACAATGGGTCGGTGTTG/CTGAATCAAGATGCCCTGAC | 58 |
| *CsLOX7* | TEA015727.1 | GGTGTTCCTGGTGCTTTCAT/CGGTCATTTCTGTAGCGATG | 59 |
| *CsLOX8* | TEA014386.1 | GACAGAAATCCGAACAGTAGGTC/ATTGCCCGAAGTTGACAGC | 59 |
| *CsLOX9* | TEA027370.1 | GGACAGAAATCCGAACGGTA/TTGCCCAAAGTTGACAGC | 58 |
| *CsLOX10* | TEA009430.1 | GGAACAAACTGTGCGACA/CGTTGGCATTGATGAGAG | 56 |
| *CsLOX11* | TEA011765.1 | GGTTATGGTGAGATTGGTGC/CAAGTGACATCAACAGGGC | 56 |
| *CsLOX12* | TEA007476.1 | GCACCAATCTCACCATAACC/CACTACTACCGAGCAGACCA | 58 |
| *CsLOX13* | TEA020832.1 | GACAGAAATCCGAACAGTAGG/GGAAAGCACATCAAGAGGG | 56 |
| *CsTBP* | HD.10G0014540 | GGCGGATCAAGTGTTGGAAGGGAG/ACGCTTGGGATTGTATTCGGCATTA | 60 |
| *CsEF-1α* | KA280301.1 | TTCCAAGGATGGGCAGAC/TGGGACGAAGGGGATTTT | 60 |

studies (*Shibata et al., 1994*; *Sinha et al., 2020*; *Podolyan et al., 2010*). The primers of *CsLOX*, *CsEF1α*, and *CsTBP* used for quantitative real-time PCR are listed in Table 1, and qRT-PCR was performed in a 20 μL total reaction system containing 10 μL of Hieff® qPCR SYBR® Green Master Mix (No Rox), 1 μL of cDNA templates, 0.8 μL of forward and reverse primers (10 μM), and 7.4 μL of ddH2O using a LightCycle® 480 Real-Time PCR System (Roche, Indianapolis, IN, USA). *CsEF1α* and *CsTBP* were selected as reference genes to normalize the data (*Zhou et al., 2018*), and thermos cycling was conducted as follows: 95 °C for 90 s, 95 °C for 5 s, 60 °C for 30 s, and 72 °C for 17 s, for 40 cycles (*Lin & Lai, 2010*). The relative expression level of genes was calculated using the $2^{-\Delta Ct}$ method (*Bogs et al., 2005*).

## Data processing and statistical analysis

Principal component analysis (PCA) and PLS-DA analysis was performed to determine the differences between metabolites of variable importance in the projection (VIP)>1 and $P < 0.05$. One-way ANOVA was performed to determine the differences among groups using SPSS (26.0) and Tukey's method. All statistical analyses were performed at either a 95% confidence interval ($P < 0.05$) or 99% confidence interval ($P < 0.01$) that was considered statistically significant.

## RESULTS

### Fatty acid profiles under different turning over intensities

A total of eight FAs were detected across all the samples with different turning over intensities (Fig. 2). Among them were three saturated fatty acids (SFAs) and five UFAs. The UFAs had higher content levels than those of SFAs across all treatments, with the former measuring 193,602.99 μg g$^{-1}$ on average and the latter 55,189.20 μg g$^{-1}$.

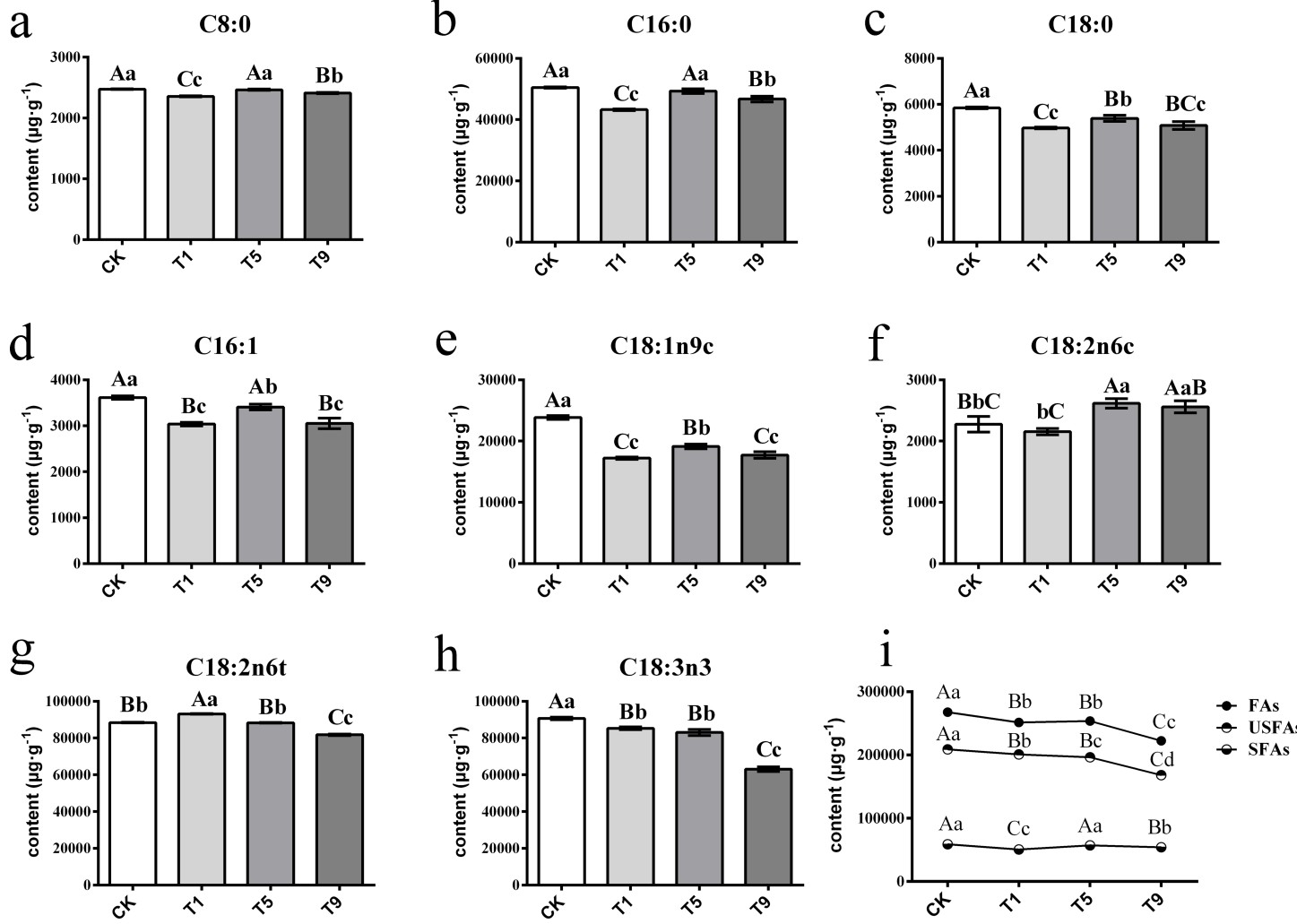

**Figure 2** **Changes of fatty acids in Tieguanyin leaves with different turning over intensities.** (A–H) were differences across eight kinds of fatty acid content in Tieguanyin leaves with different turning over intensities. (I) Content changes of total FAs, UFas, and SFAs in Tieguanyin leaves with different turning over intensities. C8:0, caprylic acid; C16:0, palmitic acid; C18:0, stearic acid; C16:1, palmitoleic acid; C18:1n9c, oleic acid; C18:2n6c, cis-linoleic acid; C18:2n6t, trans-linoleic acid; C18:3n3, linolenic acid; FAs, fatty acids; UFAs, unsaturated fatty acids; SFAs, saturated fatty acids. Different uppercase and lowercase letters represent significant differences at $P < 0.01$ and $P < 0.05$, respectively.

SFAs contained caprylic acid (C8:0), palmitic acid (C16:0), and stearic acid (C18:0), and their content ranged from 2,398.36 µg g$^{-1}$∼50,650.64 µg g$^{-1}$. C16:0 was the main component of SFAs, accounting for more than 85% of them in each sample. After the first turning over treatment (T1), the component of total tree SFAs dropped dramatically to the lowest point ($P < 0.01$), and then increased back to the level of fresh leaves in the fifth turning over treatment (T5).

UFAs contained palmitoleic acid (C16:1), oleic acid (C18:1n9c), cis-linoleic acid (C18:2n6c), trans-linoleic acid (C18:2n6t), and linolenic acid (C18:3n3), and ranged from 2,202.27 µg g$^{-1}$∼93,272.51 µg g$^{-1}$. The main components of UFAs were C18:2n6t and C18:3n3, with each accounting for more than 37% of total UFAs. Linoleic and linolenic
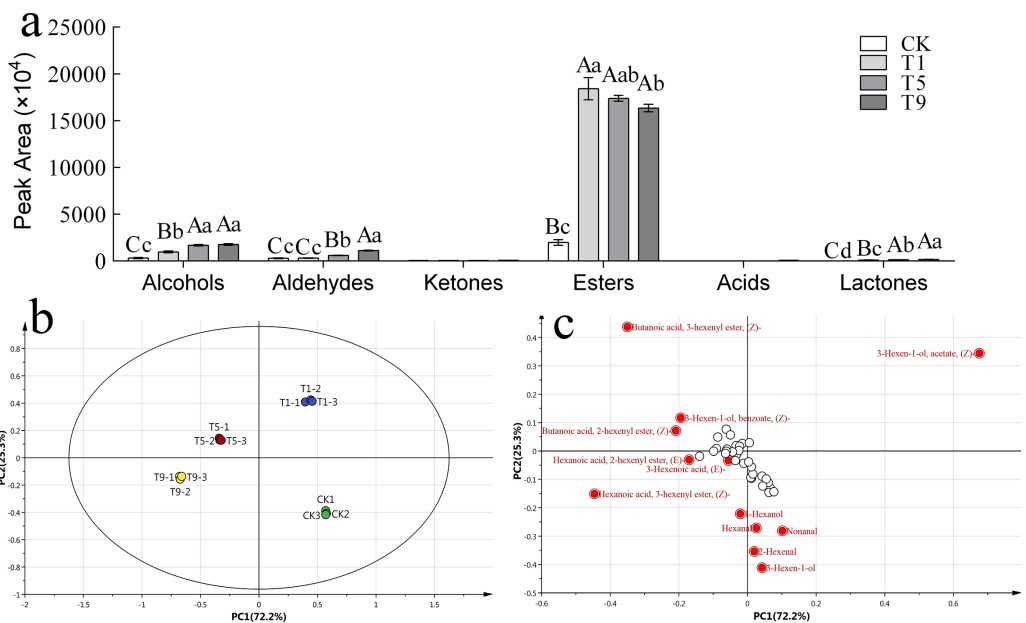

**Figure 3** Fatty acid derived volatiles in Tieguanyin leaves with different turning over intensities. (A) Peak area of fatty acid derived volatiles in Tieguanyin leaves with different turning over intensities. (B) PCA score for fatty acid derived volatiles in Tieguanyin leaves with different turning over intensities. (C) Loading-plot for fatty acid derived volatiles in Tieguanyin leaves with different turning over intensities.

acid are main reaction substrates in the fatty acid metabolic pathway for the generation of six carbon aldehydes, alcohols, and acids (*Qian et al., 2016*; *Zhang et al., 2010*). The change pattern of linoleic acid differed from its isomerism. After increasing 5.34% in T1, the content of trans-linoleic acid (C18:2n6t) kept dropping with the increase of turning over intensity. However, the content of C18:2n6c decreased slightly in T1, and then increased in T5 and T9 by 14.87% and 12.44% when compared with CK. The content of C18:3n3 decreased continuously with the increase of turning over intensity, and the highest content was 90,702.60 µg g$^{-1}$ in CK, while T1, T5, and T9 decreased by 6.08%, 8.7%, and 31.24% respectively, and T9 having the lowest of 63,082.6 µg g$^{-1}$.

Overall, the content of total FAs and UFAs dropped dramatically with the increase of turning over intensity, especially from CK to T1 and from T5 to T9, while the content of SFAs fluctuated during the whole process.

## FADV profiles under different turning over intensities

To further explore the effect of turning over intensity on volatile profiles derived from FAs, volatile compounds from all treatments were analyzed. A total of 42 types of fatty acid metabolites were detected that accounted for 17.87% of the total known volatile components, and the content of fatty acid metabolites accounted for 29.30% of the total known volatile content. The detected fatty acid metabolites could be divided into six categories: six alcohols, six ketones, eight aldehydes, 20 esters, one acid, and one lactone. The peak areas of various substances are shown in Fig. 3A.

PCA was carried out on the fatty acid metabolites of four Tieguanyin samples with different turning over intensities (Fig. 3B). According to the loading-plot figure (Fig. 3C), the four samples were dispersed in the direction of PC1 and PC2. The contribution rate of the first principal component was 72.2%, the contribution rate of the second principal component was 25.3%, and the total contribution rate was 97.5%. CK, T1, T5, and T9 are separated and can be clearly distinguished in different quadrants in the figure. CK and three turning over samples were clearly separated, indicating that the composition of fatty acid metabolites in Tieguanyin leaves changed greatly after the turning over process.

Using VIP > 1 as the screening condition, 12 volatile compounds with great contributions to the fatty acid metabolism of Tieguanyin were obtained. There were six esters that accounted for half of the characteristic volatiles, four aldehydes, and two alcohols.

According to the change trend, these volatile compounds could be divided into three categories. The first category showed a trend of first rising and then falling, and included (Z)-3-Hexen-1-ol acetate, 3(Z)-Hexenyl butanoate, and(Z)-3-Hexen-1-ol benzoate. The second category continued to rise during the turning over process, and included (Z)-3-Hexenyl hexanoate, 3-Hexen-1-ol, 2-Hexenal,1-Hexanol, (E)-2-Hexenyl hexanoate, and 2(Z)-Hexenyl butanoate. The third category showed a downward and then upward trend and included nonanal and hexanal. Interestingly, trans-3-hexenoic acid was not detected in fresh leaves nor leaves with light turning over intensity. However, a small amount of trans-3-hexenoic acid was detected in T9, and was the only acid compound in the fatty acid metabolites of Tieguanyin.

### *CsLOX* gene family expression profiles under different turning over intensities

As turning over intensities generally increased, *CsLOX* gene family members showed an upward and then downward trend (Fig. 4). Most *CsLOX* genes responded quickly to the first turning over treatment and the expression levels increased dramatically compared with CK. However, the expression level of *CsLOX13* was too low to detect. The expression levels of *CsLOX3*, *CsLOX5*, *CsLOX7*, *CsLOX8*, *CsLOX9*, *CsLOX11*, and *CsLOX12* peaked at the first turning over treatment and then dropped greatly during the fifth turning over treatment. In T1, *CsLOX5* expressed more than 83 times the level of CK. *CsLOX3 CsLOX7*, *CsLOX8*, *CsLOX9*, and *CsLOX11* also expressed more than five times after the first turning over treatment compared with CK. However, the expression level of *CsLOX1* increased continuously from CK to T9, and the expression level in T9 was 29 times and three times higher than that in CK and T1, respectively. *CsLOX2, CsLOX4, CsLOX6,* and *CsLOX10* showed a similar pattern in their expression levels between the first and fifth turning over treatments and maintained a relatively high expression level before T9. Overall, *CsLOX* gene family members hardly expressed in CK, although they could be triggered by immediate turning over treatment stress. They either remained at a relatively high expression level or dropped with the increase of turning over intensities, depending on the different *CsLOX* members.

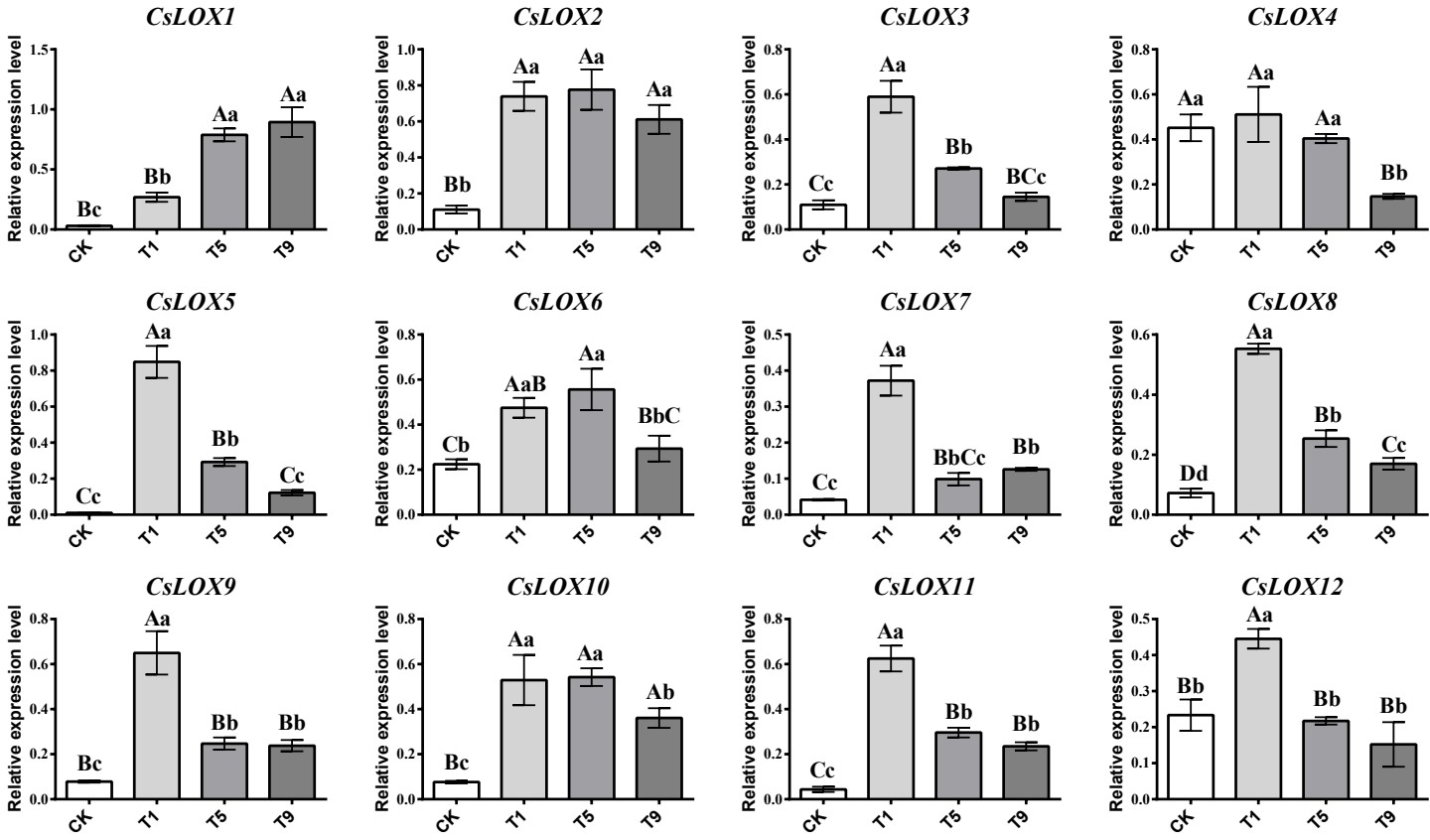

**Figure 4** **Relative expression level of *CsLOX* gene family members in Tieguanyin leaves under different turning over intensities.** Different uppercase and lowercase letters represent significant differences at $P < 0.01$ and $P < 0.05$, respectively.

## Correlation analysis of fatty acid substrates, FADVs, and *CsLOX* genes

Most *CsLOX* gene family expression levels showed a negative correlation with fatty acid substrates and a positive correlation with key FADVs (Fig. 5). *CsLOX1* had a significantly negative correlation with UFAs and FAs ($P < 0.01$), and significantly positive correlations with all FADVs shown in Table 1 except for (Z)-3-Hexen-1-ol acetate. The Pearson correlation coefficient of *CsLOX1* with 1-Hexanol, 3-Hexen-1-ol, (Z)-3-Hexen-1-ol benzoate, 2(Z)-Hexenyl butanoate, (E)-2-Hexenyl hexanoate, and (Z)-3-Hexenyl hexanoate were all higher than 0.9 ($P < 0.01$). *CsLOX2* also showed a significantly positive correlation with ester volatiles, as well as *CsLOX10*. However, *CsLOX4* and *CsLOX12* showed different patterns with other *CsLOX* members; it showed negative correlations with the volatiles 1-Hexanol, 2-Hexenal, hexanal, nonanal, and (E)-2-Hexenyl hexanoate. (Z)-3-Hexen-1-ol acetate, which has a strong fruit aroma, was significantly positively correlated with the expression of *CsLOX3*, *CsLOX5*, *CsLOX7*, *CsLOX8*, *CsLOX9*, *CsLOX10*, *CsLOX11*, and *CsLOX12* with Pearson correlation coefficients of 0.986, 0.991, 0.935, 0.992, 0.963, 0.767, 0.981, and 0.833 respectively ($P < 0.01$). (Z)-3-Hexen-1-ol acetate also significantly positively correlated with the expression of *CsLOX2* and *CsLOX6* with

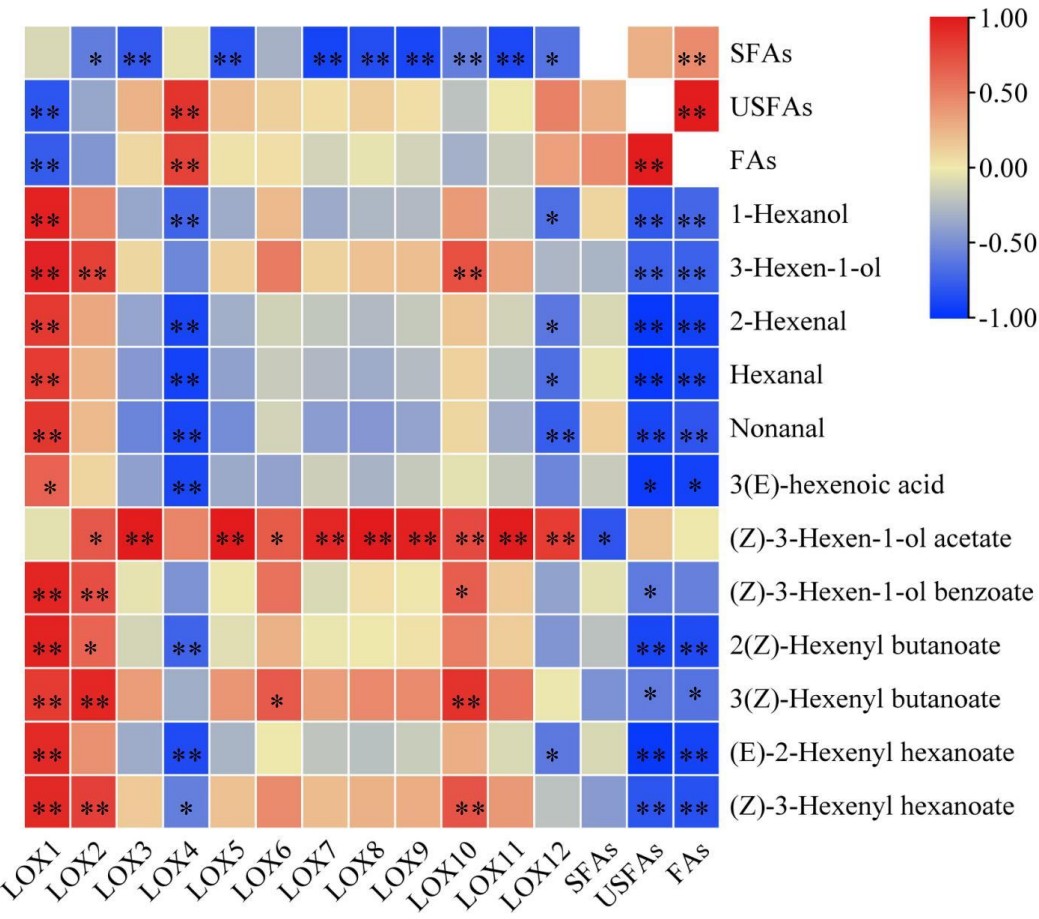

**Figure 5** Correlation analysis of fatty acid substrates, fatty acid derived volatiles, and *CsLOX* genes. An asterisk (*) and two asterisks (**) and represent significant differences at $P < 0.05$ and $P < 0.01$, respectively.

Pearson correlation coefficients of 0.693 and 0.687, respectively ($P < 0.05$). Furthermore, UFAs showed significant correlations with all the shown volatiles except (Z)-3-Hexen-1-ol acetate, and SFAs showed significant correlations with all the shown volatiles except (Z)-3-Hexen-1-ol acetate and (Z)-3-Hexen-1-ol benzoate.

## DISCUSSION

### Fatty acid content in different turning over treatments

In our study, the total fatty acid content decreased dramatically with the turning over process, and the decrease of UFA content played the most important role. This indicates that the heavier the turning over intensity, the more the fatty acid in tea leaves is degraded. In Tieguanyin oolong tea, the top three FAs were linolenic acid (C18:3n3), trans-linoleic acid (C18:2n6t), and stearic acid (C16:0), which was a similar result to those of previous studies (*Guo et al., 2019*). Continuous mechanical stress could cause the degradation of linoleic acid (*Zeng et al., 2018*). Notably, we observed all three UFAs showed a similar trend

where the content decreased first in , rose back in T5, and finally dropped again in T9. The mechanical stress in T1 might be the major reason for the degradation of UFAs, but the rise in T5 was probably caused by the transformation from SFAs to UFAs. We also noticed an increase of cis-linoleic acid during the turning over process, which may suggest that some trans-linoleic acid changes to cis-linoleic acid during the process.

## FADVs in different turning over treatments

Metabolites such as aroma can be greatly triggered by abiotic stress during tea processing (*Wang et al., 2010*; *Yang & Liang, 2008*). After turning over treatment, the content of FADVs increased. The detected fatty acid metabolites were divided into six categories: alcohols, ketones, aldehydes, esters, acids, and lactones. The ester compounds with pleasant flower aromas were the most abundant, and the kinds of alcohols and aldehydes with fresh flower and fruit aromas were also relatively high. The PCA results showed that the components of fatty acid metabolites in postharvest leaves changed greatly after turning over treatment, which indicated that turning over treatment has a great effect on fatty acid metabolism in postharvest leaves.

Using VIP > 1 as a screening condition, 12 volatile compounds were obtained. (Z)-3-Hexenyl hexanoate usually presents with a fresh fruit aroma (*Liu et al., 2019*), and its content continued to rise and reached its peak when it was intensely turning over. (Z)-3-Hexen-1-ol acetate with a strong fruit and banana-like aroma increased first and then decreased, and the content was the highest when the leaves were slightly turning over. 3(Z)-Hexenyl butanoate with a fresh fruit-like aroma increased to its highest point when it was moderately turning over. Green tea, which does not undergo the turning over process, has a fresh aroma, while oolong tea has abundantly fruity scent (*Wang, 2020*; *Ji et al., 2016*). As the turning over intensity increases, the aroma might change from a fresh leaf-like smell to a fresh fruit-like one, and finally to a sweet fruit-like smell. According to research on total volatile compounds and volatile compounds in the fatty acid metabolic pathway, Tieguanyin has rich volatile compounds and high content during moderate turning over (T5), and the content of main metabolites of FAs is high. Therefore, we suggest that during production, Tieguanyin with medium turning over intensity may obtain high content, rich kinds of volatile compounds, and better aroma quality.

## Correlation analysis of fatty acid substrates, volatiles, and genes

The correlation analysis of fatty acid substrates (SFAs, UFAs, and FAs), key FADVs (VIP > 1), and the expression level of *CsLOX* genes was performed. It is notable that (Z)-3-Hexen-1-ol acetate, which has a fruit-like aroma, showed significant positive correlations with 10 out of 12 *CsLOX* members. The *CsLOX* family could work together when performing their functions. Previous studies showed that the oxidative degradation of fatty acids might be subject to stresses such as mechanical force and water loss, and mainly produce $C_6$ aldehyde, $C_6$ alcohol, and leaf alcohol esters. Continuous mechanical damage could activate the *CsLOX* gene, and the main products contain hexenal, hexenol, and leaf alcohol esters (*Zeng et al., 2018*; *Zhou et al., 2021b*). It was also reported that the protein encoded by *CsLOX1* is a kind of 9/13-LOX protein with double cleavage sites that belongs not

only to the 9-LOX subfamily and is different from the other *LOX* gene family members (*Zhou et al., 2017a*; *Zhou et al., 2017b*). In this study, we noticed that the *CsLOX1* relative expression level increased during the turning over process and was higher than those of most of the other *LOX* genes, which was a similar result to that of *Zhou et al. (2017a)* and *Zhou et al. (2017b)*. *Zeng et al. (2018)* also found that the expression level of *CsLOX1* was significantly enhanced by the continuous mechanical damage. *CsLOX1* had a significantly negative correlation with UFAs and FAs, and it had a significant positive correlation with all the key volatiles except (Z)-3-Hexen-1-ol acetate. We expect that *CsLOX1* plays a main role in the degradation of fatty acid substrates and the performance of key FADVs, but further studies are needed to confirm our speculation.

## CONCLUSION

In this study, we present a whole profile of FA metabolites during the turning over part of oolong tea's manufacturing process. Our FA analysis revealed that C16:1, C18:1n9c, C18:2n6t, and C18:3n3 might be key substrates in the fatty acid metabolic pathway because they dropped as the turning over intensity and FADVs increased. Furthermore, we identified a key fruit-like aroma, (Z)-3-Hexen-1-ol acetate, which had strong positive correlation with most *CsLOX* genes. The expression level of *CsLOX1* rose dramatically with the increase of turning over intensities, and it was considered a main contributor to the formation of fatty acid derived aromas. We observed a possible pattern regarding aroma formation and changes during the turning over process from the perspective of the fatty acid pathway and found that Tieguanyin had a relatively abundant and pleasant aroma under moderate (five times) intensity. However, how other metabolic pathways, form, change, and contribute to the whole aroma quality of oolong tea requires further exploration. Our findings could help improve understanding of the mechanism and effect of different turning over treatments, as well as the quality of oolong tea and its manufacturing process.

### Funding
This research was funded by the China Agriculture Research System of MOF and MARA: The Earmarked Fund for China Agriculture Research System, grant number CARS-19, Science and Technology Innovation Project of Fujian Agriculture and Forestry University, grant number CXZX2017177. The funders had no role in study design, data collection and analysis, decision to publish, or preparation of the manuscript.

### Grant Disclosures
The following grant information was disclosed by the authors:
China Agriculture Research System of MOF and MARA: The Earmarked Fund for China Agriculture Research System.
Science and Technology Innovation Project of Fujian Agriculture and Forestry University: CXZX2017177.

## Competing Interests

The authors declare there are no competing interests.

## Author Contributions

- Zixin Ni conceived and designed the experiments, performed the experiments, analyzed the data, prepared figures and/or tables, authored or reviewed drafts of the article, and approved the final draft.
- Qingyang Wu conceived and designed the experiments, performed the experiments, analyzed the data, prepared figures and/or tables, authored or reviewed drafts of the article, and approved the final draft.
- Ziwei Zhou conceived and designed the experiments, performed the experiments, prepared figures and/or tables, authored or reviewed drafts of the article, and approved the final draft.
- Yun Yang performed the experiments, authored or reviewed drafts of the article, and approved the final draft.
- Qingcai Hu performed the experiments, authored or reviewed drafts of the article, and approved the final draft.
- Huili Deng performed the experiments, authored or reviewed drafts of the article, and approved the final draft.
- Yucheng Zheng performed the experiments, authored or reviewed drafts of the article, and approved the final draft.
- Wanjun Bi performed the experiments, authored or reviewed drafts of the article, and approved the final draft.
- Zhenzhang Liu performed the experiments, authored or reviewed drafts of the article, and approved the final draft.
- Yun Sun conceived and designed the experiments, authored or reviewed drafts of the article, funding acquisition, Project administration,, and approved the final draft.

## Data Availability

The raw data of volatiles are available in the Supplementary File.

## Supplemental Information

Supplemental information for this article can be found online at http://dx.doi.org/10.7717/peerj.13453#supplemental-information.

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
