# Peer review of "Effects of turning over intensity on fatty acid metabolites in postharvest leaves of Tieguanyin oolong tea (Camellia sinensis)"

_PeerJ, doi:10.7717/peerj.13453_

## Round 0.1 · original submission · Major Revisions

The manuscript presents interesting results on tea plant biology. However, there are a lot of critical remarks from the reviewers to fix. Please check the comments from reviewer #1.

Reviewer 1 ·

Basic reporting

1. There are some grammatical mistakes. It is suggested that we should standardize our own words, I strongly recommend that the authors should check the manuscript more thoroughly and ask fluent English speaker to edit their manuscript.

2. The format is chaotic, either texts or reference. For instance, non-uniform units, non-standard Latin names, casual typeset, and un-regulated superscript and subscript. Please check them through the manuscript.

3. Introduction section should give more details for readers. Especially, give more details of new advances in research on fatty acids in Camellia sinensis. The literature regarding LOX genes cited in the introduction is older, please add more relevant literatures support for the past three years.

Experimental design

1. Materials & Methods section Is there any reference or pre-test for this treatment method, that is, whether this method is reasonable or not, please add reference or pre-test data.

2. Statistical analysis is missing. It needs to be supplemented in Figure 3A.

3. The image of Figure 3C is blurred and can't be seen clearly. It is suggested to adjust and modify Figure 3C and other unclear pictures.

Validity of the findings

1. The results of LOX genes in Camellia sinensis are inconsistent. According to the table 1, 13 LOX genes were mined, however, in Figure 4 and 5, only 12 LOX genes were investigated. where is CsLOX13? These discrepant results should be better illustrated.

2. The authors declared that “CsLOX1 might be the main gene to play an important role in the degradation of fatty acid substrates and the performance of key fatty acid derived volatiles”. It is an interesting speculation. More detailed evidences are required.

3. Discussion section “Techniques in investigating volatiles” is incomplete and deficient. More information is required. Furthermore, what is the link between “Techniques in investigating volatiles” and other topics?

Additional comments

Manuscript ID: 68064
Manuscript title: Effects of turning over intensity on fatty acid metabolites in postharvest leaves of ‘Tieguanyin’ oolong tea (Camellia sinensis)

In present manuscript, the authors investigated the effects of turning over intensity on fatty acid metabolites, including total fatty acids, saturated fatty acids, unsaturated fatty acids, and characteristic fatty acid derived volatiles, in postharvest leaves of ‘Tieguanyin’ oolong tea (Camellia sinensis). Overall, it has brought some interesting results. However, the following concerning should be also addressed.

The results of LOX genes in Camellia sinensis are inconsistent. According to the table 1, 13 LOX genes were mined, however, in Figure 4 and 5, only 12 LOX genes were investigated. where is CsLOX13? These discrepant results should be better illustrated.

The format is chaotic, either texts or reference. For instance, non-uniform units, non-standard Latin names, casual typeset, and un-regulated superscript and subscript. Please check them through the manuscript.

Statistical analysis is missing. It needs to be supplemented in Figure 3A.

Introduction section should give more details for readers. Especially, give more details of new advances in research on fatty acids in Camellia sinensis. The literature regarding LOX genes cited in the introduction is older, please add more relevant literatures support for the past three years.

Materials & Methods section Is there any reference or pre-test for this treatment method, that is, whether this method is reasonable or not, please add reference or pre-test data.

The image of Figure 3C is blurred and can't be seen clearly. It is suggested to adjust and modify Figure 3C and other unclear pictures.

Discussion section “Techniques in investigating volatiles” is incomplete and deficient. More information is required. Furthermore, what is the link between “Techniques in investigating volatiles” and other topics?

The authors declared that “CsLOX1 might be the main gene to play an important role in the degradation of fatty acid substrates and the performance of key fatty acid derived volatiles”. It is an interesting speculation. More detailed evidences are required.

Reviewer 2 ·

Basic reporting

Although the fatty acid derived volatiles are one types of key aroma compounds in oolong tea and had been well studied, in this manuscript, Ni and co-authors detected the changes of fatty acids, their derived volatiles and LOX genes’ expression patterns during oolong tea’s turn over processing. The experiments are well conducted and interpreted. The results of this study will provide a valuable information to understand the aroma formation during oolong tea processing.

Experimental design

1. How to perform the turn over processing?
2. How to define the turn over intensity? From the methods, T1, T5 and T9 is the different time points during oolong tea, they have no relationship with intensity? So I suggest the authors to change the title and “intensity” using “degree/stage of turn over processing”.
3. How to give the names of CsLOX genes, the reference should be listed.

Validity of the findings

4. Line 302-310, delete.
5. The name of genes should be expressed as italic, such as in the title of Fig. 4.

---

## Round 0.2 · Minor Revisions

Thanks for the manuscript update. The reviewers have no more critical comments. However due to the recent revision, some parts of the text have typos. It should be fixed before publication. Some of the figures should be improved.

Please check the version with tracked changes. It has inserted comments not relevant to the text (delete it). Due to multiple editing there are some new typos to be checked.

Delete redundant abbreviations in the Abstract.
Line 24: (FAs) - delete wording - it is not used in the Abstract again, not need here
Line 26: (UFAs) - delete.

Line 30: ‘considered to be a leader’ - please rephrase, use words ‘key gene’, it is not proper phrase to use word ‘leader’.
‘Correlation analysis showed that’ - this phrase could be rewritten too. Word ‘correlation’ used twice in the same sentence.
Just write ‘We found that’ instead of ‘correlation analysis’.
Line 31: ‘CsLOX genes.’ - may comment here about the abbreviation - add ‘Camelia sinensis LOX family genes’.

Correlation is with gene expression, not with genes itself. Please use correct phrase.
Line 32: ‘moderate (five times)’ - please add details what means 5 times?
PeerJ is open journal for all the biologists, a reader might be not familiar with the tea processing standards. I see proper detail what is ‘turning over’ on lines 43 and down in the main text.

Please check writing ‘min -1’ without extra spaces (‘ ’). Lines 131-133.
Same remark for the measurement units (temperature, eV)
Line 188: ‘g -1’ - keep it without spaces on the same line. Adjust spaces and formatting to keep digits and measurement units on the same line

Line 359, 362 - check spaces sign before comma ( ,) to keep proper formatting in the references.
Line 394: ‘XuXQ’ - wrong name spelling
Line 397: - dots without spaces
Line 457: ‘Zhou, ZW’ - it mixes formatting, should be uniform without comma.
Overall, please check formatting in the updated parts of the text.

Figure 3 panel (c) - please increase the font. Or remove the compound names in black around dots.
Table 1 - format the line for the primer AT1G55520 in one row.

Reviewer 3 ·

Basic reporting

The study provides an overall view of how fatty acid metabolites change and affect the quality of oolong tea with different turning over intensities during processing. The manuscript is well-written after the revision.

Experimental design

The Experimental Design of the manuscript is reasonable and the method is described in detail.

Validity of the findings

The fingdings are of great significance in guiding the control of tea aroma quality.

Additional comments

The manuscript is suggested to be accepted for publication in the version.

---

## Round 0.3 · accepted · Accept

Thanks for the update and answer to the editor. There are no more critical comments on the text.